# DISENTANGLED REPRESENTATIONS OF MICROSCOPY IMAGES

## ABSTRACT

Microscopy image analysis is fundamental for different applications, from diagnosis to synthetic engineering and environmental monitoring. In the last few years, the number of available images has been constantly growing, thanks to technological advancements, pushing toward the development of automatic image analysis methods based on deep learning. Although deep neural networks have demonstrated great performance in this field, interpretability — an essential requirement for microscopy image analysis — remains an open challenge. This work proposes a Disentangled Representation Learning (DRL) methodology to enhance model interpretability for microscopy image classification. Exploiting benchmark datasets coming from three different microscopic image domains, including plankton, yeast vacuoles, and human cells, we show how a DRL framework, based on transfer learning from synthetic features, can provide a good trade-off between accuracy and interpretability in this domain.

## 1 INTRODUCTION

The analysis of microscopy images is crucial to biomedical research (Liu et al., 2021), from histopathology diagnostic to cell organelles analysis and microorganisms classification. Manual analysis has become impractical in recent years, given the huge volumes of images acquired through advanced acquisition systems (Pastore et al., 2023b). Consequently, deep learning has been extensively applied to the microscopy domain for different image analysis tasks (Xing et al., 2017), including classification (Liu et al., 2022a; Fırat, 2024), segmentation (Greenwald et al., 2022; McKinley et al., 2022) and object detection (Rivas-Villar et al., 2020; Hung et al., 2020; Kumar et al., 2023). On the one hand, Deep Neural Networks (DNNs) have been reaching great performance for microscopy images related tasks, generally outperforming conventional approaches based on hand-crafted features (Plissiti et al., 2018; Pastore et al., 2023a). On the other hand, given the intrinsic DNN's *black box* nature, predictions lack interpretable, human-reliable insights, which are highly desirable in this specific domain of application Krishna et al. (2023); Tavolara et al. (2023). Consequently, providing interpretable deep-learning methods for biological images is a pressing research challenge (Bera et al., 2019).

In this work, we propose a Disentangled Representation Learning (DRL) framework (Bengio et al., 2013; Locatello et al., 2019b; Higgins et al., 2017a; Wang et al., 2023) as a potential method to enhance DNN's interpretability in this context. DRL aims to learn models that can identify and disentangle underlying Factors of Variation (FoVs), hidden in the observable data, encoding them in an *interpretable* and compact way, partially independently from the task at hand enhancing robustness, and generalization capacity across various tasks (Kulkarni et al., 2015; Chen et al., 2016; Bengio et al., 2013; Zhu et al., 2021; Locatello et al., 2019a; Van Steenkiste et al., 2019; Wang et al., 2023). DRL was studied in depth in (Chen et al., 2016) and (Higgins et al., 2017b), and following these works many attempts have been made to learn effectively disentangled representations. Kahana & Hoshen (2022) propose a contrastive learning paradigm, while the work in Lin et al. (2020) they present a contrastive regularization for disentangled GANs. The authors in Song et al. (2024) introduce Flow Factorized Representation Learning which defines a set of latent flow paths that correspond to sequences of different input transformations that resemble the FoVs, while in Ren et al. they leverage pretrained generative models for discovering traversal directions as factors with contrastive learning. The authors in Yang et al. (2024) disentangle the gradient fields of the Diffusion

Probabilistic Models to discover factors automatically, but finding that the representation may not be easily interpretable by humans.

While the earliest approaches face disentanglement in an unsupervised fashion, without any explicit definition of the Factors of Variation (FoVs), it has been shown that weakly-supervised approaches to disentanglement provide superior results, see for instance Ada-GVAE (Locatello et al., 2020; Fumero et al., 2021). However, they find limited applicability due to the general lack of knowledge on FoVs characterizing real data. Transfer learning offers a suitable solution, by transferring a disentangled representation from a Source dataset – where FoVs are known – to a Target one – where FoVs might be unknown. In this direction, a recent study (Dapueto et al., 2024) shows that a disentangled representation learnt from a synthetic dataset can be transferred to a real one preserving a partial level of disentanglement. Although promising, the analysis is limited to real dataset whose FoVs are controlled and known a priori.

In this work, we move a step further in assessing the applicability of weakly supervised DRL to real-world tasks, by addressing the specific problem of enhancing interpretability of single cell microscopy image classification, where the FoVs are only partially known. On the methodology side, this domain allows us to deal with a real task where the number of possible FoV is limited, compared with conventional pictorial images. On the application side, we propose an automatic procedure to learn interpretable representations in a field where interpretability is as important as accuracy. Specifically, we adopt datasets coming from three different biology domains: plankton microorganisms, budding yeast vacuoles, and human cancer cells. The datasets we adopt differ in acquisition systems and type of imaged cell but can be semantically described in terms of relatively simple morphological factors (e.g. texture, shape, color, scale, and other morphological features), thus offering a perfect benchmark for our scope. At the same time, microscopy datasets (Pastore et al., 2020; Ciranni et al., 2024; Sosik et al., 2015), like most real-world data, are not associated with any specific FoV annotation. To overcome this problem, we take inspiration from (Dapueto et al., 2024) and transfer a latent representation from an Ada-GVAE trained on a different Source dataset to a Target microscopy dataset. As a source, we adopt a simple synthetic dataset, Texture-dSprite, whose (annotated) FoVs may be appropriate to represent the morphological factors we are interested in. As for the interpretation of the results we exploit the availability in the literature of several hand-crafted features computed on the same datasets, that we use as a reference, computing the correlation between these features and the learned dimensions. In this sense, our work aims to learn the FoVs of the microscopy data following a fully data-driven approach instead.

Moreover, inspired by recent works on unsupervised learning from biological image datasets (Pastore et al., 2023b), instead of using the images directly as inputs of Ada-GVAE, we provide a projection into a large-dimensional vector of deep features, obtained by a ViT16b model pretrained with DINO self-supervised approach on ImageNet. With our analysis, we demonstrate that our approach achieves not only good classification accuracy but also disentanglement performances comparable to those learned with synthetic datasets, thus enhancing the interpretability of the learned representations.

To summarize, the main contributions of this paper are the following:

- We assess a weakly-supervised Disentangled Representation Learning procedure on real datasets, with unknown or partially known FoVs. We focus on microscopy images, where disentanglement is a desired property, the data are complex, but the FoV are somewhat controlled in number.
- We adopt an input representation based on pre-trained deep features in the DRL framework. Our results on microscopy image datasets show that such design choice allows us to significantly increase the classification accuracy of our interpretable DRL framework with respect to the usage of plain images.

To the best of our knowledge, this work represents the first application of DRL to real-world datasets and the first attempt of learning the disentangled representations from pretrained features.

We believe this paper could serve as a foundation for integrating interpretability derived from disentanglement into deep learning frameworks for microscopy image classification tasks, and more broadly, for real-world applications. The remainder of the paper is organized as follows: in Section 2 we describe the proposed DRL methodology, in Section 3 we describe the microscopy image

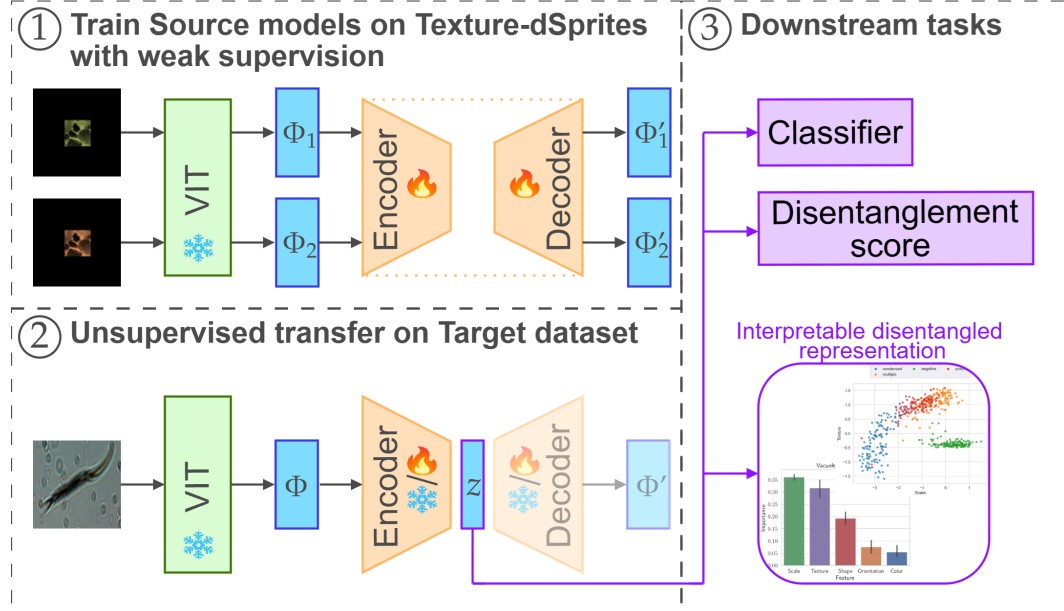

Figure 1: A visual sketch of our methodology. We project each image into a high-dimensional deep feature vector using a pretrained network (ViT16b model pretrained with DINO self-supervised approach on ImageNet1K). Then, our approach includes 3 main components. (1) We learn a disentangled model with weak-supervision using Ada-GVAE, using a Source annotated dataset (Texture-dSprite in this case). (2) Then, we transfer the pretrained disentangled model to a real Target dataset (microscopy images in this work). (3) We evaluate the quality of the disentangled model and of the corresponding representation using disentanglement scores and the classification accuracy in a downstream task (the one associated with the Target dataset).

datasets used in this work, provide implementation details, and report the obtained results, evaluating interpretable insights for each dataset, finally discussed in Section 4.

## 2 THE PROPOSED MEDOLOGY

Although the majority of existing methods may be based on different definitions of disentanglement (see for instance (Bengio et al., 2013; Higgins et al., 2018; Suter et al., 2019)) there is a general agreement that, not surprisingly, disentanglement can be better achieved with some level of supervision on the FoVs (Locatello et al., 2019b). However, in our target scenario, i.e. microscopy real data, there is no availability of benchmark datasets with this type of annotation.

Therefore, in this work we exploit a transfer learning paradigm, to transfer a disentangled model – trained with weak supervision on a Source dataset where annotation of FoVs is available – to a Target real dataset – where we may assume FoVs to be present in the data but they are unknown. To this purpose, we follow the methodology proposed in Dapueto et al. (2024), according to the pipeline in Fig. 1.

### 2.1 TRANSFERRING DISENTANGLED REPRESENTATIONS TO REAL DATA

In contrast to the large majority of previous works, instead of disentangling the features extracted from the raw images, we first project the images into a large vector of deep features $\Phi$, by using a pretrained network. Although different choices are possible, we empirically observed that in our scenarios the ViT16b model pretrained with DINO self-supervised approach on ImageNet1K (Caron et al., 2021) is the most appropriate since it better captures the complexity of the Source data (see the comparison in Appendix A.2.1). This choice has been inspired by recent works on microscopy image analysis, showing that such models provide rich and discriminative features for the task at hand, (Pastore et al., 2023b; Lumini et al., 2020; Kyathanahally et al., 2022), generally outperform-

Table 1: Balanced accuracy (%) and standard deviation of the classifiers trained on the disentangled representation extracted from the VAE (Lensless dataset) .

| Source | Input | ✗ Finetuning | | ✓ Finetuning | |
|--------|-------|--------------|--------------|--------------|--------------|
| | | GBT | MLP | GBT | MLP |
| T.-dSpr. | RGB | $70.32 \pm 0.029$ | $71.93 \pm 0.030$ | $73.04 \pm 0.024$ | $75.48 \pm 0.027$ |
| T.-dSpr. | $\Phi$ | $77.06 \pm 0.020$ | $77.46 \pm 0.022$ | $93.55 \pm 0.019$ | $94.62 \pm 0.017$ |

ing models trained from scratch (Ren et al.).

Following the same procedure as in Dapueto et al. (2024), we derive the latent disentangled representation using Ada-GVAE (Locatello et al., 2020) on the Source dataset with annotated FoVs. More specifically, the input of the VAE is a pair $\Phi(\boldsymbol{x}_1)$ and $\Phi(\boldsymbol{x}_2)$ for which the images $\boldsymbol{x}_1$ and $\boldsymbol{x}_2$ are sampled from the dataset so that they differ of $k$ FoVs, with $k$ fixed.

Then, we transfer the representation on the Target datasets by finetuning the models on the real unsupervised data with $\beta$-VAE (Higgins et al., 2017b). We adopted Ada-GVAE and $\beta$-VAE to allow a fair comparison with Dapueto et al. (2024) highlighting that the main difference is in the choice of the input – we adopt the deep features $\Phi$ produced by DINO instead of the the RGB images proposed in the previous approach.

## 2.2 DISENTANGLEMENT EVALUATION METHODS

For a quantitative evaluation of disentanglement, there is common agreement on the fact that a disentangled representation should satisfy the following properties (Do & Tran, 2020; Van Steenkiste et al., 2019; Bengio et al., 2013). **Modularity**: a factor influences only a portion of the representation space, and only this factor influences this subspace (Ridgeway & Mozer, 2018; Eastwood & Williams, 2018). **Compactness** or completeness: the subset of the representation space affected by a FoV should be as small as possible (ideally, only one dimension) (Ridgeway & Mozer, 2018). **Explicitness**: DR should explicitly describe the factors, thus it should allow for an effective FoVs classification (Ridgeway, 2016).

As shown in Fig. 1, right, we evaluate the quality of disentanglement in our transfer learning scenario by analysing the disentanglement score and the accuracy of a downstream classification task (whose formulation depends on the specific real Target dataset). Among the several disentanglement scores from the literature we consider a selection that allows us to capture the different properties mentioned above. More specifically, our analysis includes DCI, measuring Modularity (Eastwood & Williams, 2018), MIG, evaluating Compactness (Chen et al., 2018), and OMES incorporating both Modularity and Compactness (Dapueto et al., 2024). The latter also facilitates the interpretability of the results, and for this reason well suits our needs. The downstream classification task allows us to evaluate the descriptive power of the representation (explicitness). To this purpose, we rely on a simple classifier that does incorporate any further representation learning step, to evaluate the properties of the disentangled representation.

## 3 EXPERIMENTAL ANALYSIS

### 3.1 DATASETS

REAL TARGET DATASETS

**Plankton Lensless** microscope dataset (Pastore et al., 2020) (Fig. 8) consists of images acquired using a lensless microscope, extracted from 1-minute videos. It includes 10 classes, with 640 color images each. The dataset includes precise binary masks of each sample so that is possible to suppress the background and compute handcrafted features like scale, shape, and mean color. Train-test splits are provided.

**Plankton WHOI15** (Ciranni et al., 2024) (Fig. 9) is a subset of the WHOI dataset (Sosik et al., 2015), including 15 classes acquired among 4 years of acquisition (2007-2010). In our experiments, we select the year 2007 subset. With respect to Lensless, this dataset is more challenging due to its fine granularity with high intraclass variability. All the images are grayscale with varying sizes.

Segmentation masks, hand-crafted features, and train-test splits are not available. We build the test set by randomly selecting 20% of images maintaining a balanced number of samples per class.

**Budding yeast vacuoles** (Pastore et al., 2023a) (Fig. 10) dataset includes a total of 998 fluorescence budding yeast vacuole images, extracted from acquired 3D stacks, already divided into training (775) and test (205) sets. Each 3D data volume is converted into a 2D projection in which depth is encoded by color. The dataset is labelled into 4 different morphotypes: single vacuole, multiple, condensed, and negative (or dead cells). A set of hand-crafted shape- and texture-based features are available for this dataset (see (Pastore et al., 2023a))

**Sipakmed Human Cells** (Plissiti et al., 2018) dataset (Fig. 11) consists of 4049 images of isolated cells that have been manually cropped from 966 cluster cell images of Pap smear slides. These images were acquired through a camera adapted to an optical microscope. The cell images are divided into five categories including normal, abnormal and benign cells. A set of hand-crafted features describing the nucleus and cytoplasm is available for this dataset (see Plissiti et al. (2018)).

SOURCE DATASET

To facilitate the transfer of a disentangled model from a Source to a Target dataset, it is worth considering the general properties of the target data, since these may give useful insights on the possible semantics of the FoVs. Indeed, as previously mentioned, in our methodology we assume the Target data to be characterized by the presence of underlying FoVs but without a full knowledge of them nor the availability of an annotation. A common trait of the microscopy datasets described above is the possibility of computing hand-crafted morphological features including size, scale, etc.. Hence, as a Source dataset, we adopted a dataset we generated for the purpose, we name **Texture-dSprites**. It is an annotated synthetic dataset, obtained as an extension of dSprite (Matthey et al., 2017) by adding 5 textures from Abdelmounaime & Dong-Chen (2013) to the geometric shapes (3 possibles values) included in the original dSprite. In addition to `Texture` and `Shape`, the dataset includes 7 different `Colors`, 6 values of `Scale` and 40 values for the `Orientation`. The original dSprites also includes 32 x-positions and 32 y-positions. However, since in our target datasets the object of interest is always centred these factors can be neglected. Table 5 reports the dataset FoVs and some random samples.

### 3.2 IMPLEMENTATION DETAILS

Since the images in all datasets are of different dimensions, we padded them to preserve the original aspect ratio and then resized them to $224 \times 224$.

We train Ada-GVAE on the synthetic Source Texture-dSprite dataset, using pairs of images that differ in $k = 1$ factors of variation according to Locatello et al. (2020), where this was shown to lead to higher disentanglement. Following Dittadi et al. (2020); Dapueto et al. (2024), we vary the parameter $\beta$ in $\{1, 2\}$. We produce 20 Source models (10 random seeds $\times$ 2 values of $\beta$, latent dimension 10 for all) with the Adam optimizer (Kingma & Ba, 2014) and default parameters, batch size=64 and 400k steps. We use linear deterministic warm-up (Dittadi et al., 2020; Sønderby et al., 2016; Bowman et al., 2015) over the first 50k training steps. For the unsupervised finetuning on the microscopy dataset with $\beta$-VAE, we finetuned the model for 20 epochs.

### 3.3 EVALUATION PROTOCOL

We summarise the key elements of our experimental analysis, whose aim is to assess the potential of DRL of microscopy images, reported in the remainder of the section. We start from the disentangled representation $z$ (see Fig. 1) and remove the "inactive" dimensions with a standard deviation below a certain threshold.

**Downstream task performances.** To assess the efficacy of our representation with respect to the microscopy classification downstream tasks, we adopted two simple classifiers, a Gradient Boosted Trees (GBT) (Friedman, 2001) and a Multilayer Perceptron (MLP) (Lippmann, 1994) with 2 hidden layers of size 256, to better appreciate the influence of the representation on the performance in the downstream task. On the specific choices, we referred to (Dittadi et al., 2020). We analyse the results without and with finetuning (marked with ✗ and ✓, in the tables). The mean scores and standard deviation over the 20 models are reported. Classification is evaluated using balanced accuracy.

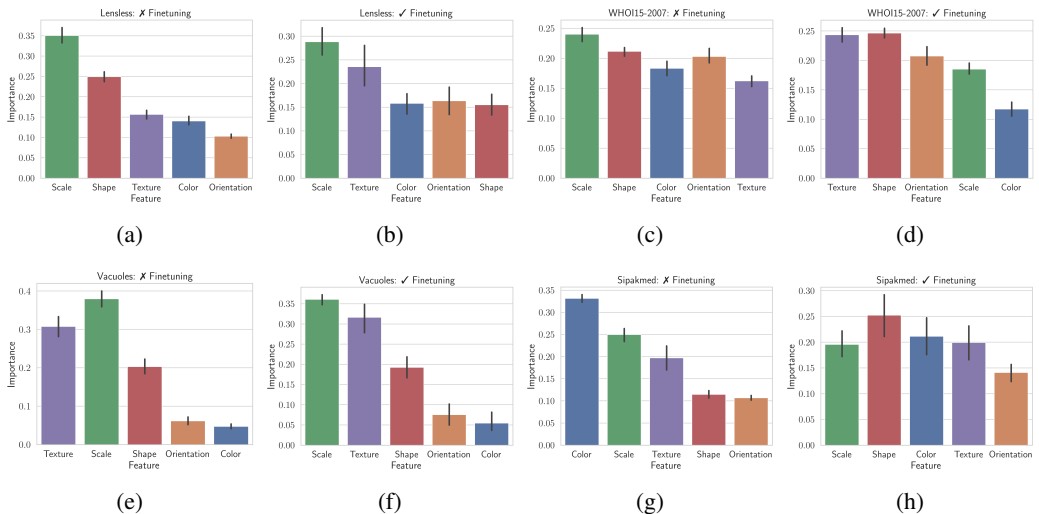

Figure 2: The mean and standard deviation of the feature importance without and with finetuning on Lensless (Fig. 2a and 2b), WHOI15 (Fig. 2c and 2d), vacuoles (Fig. 2e and 2f) and Sipakmed (Fig. 2g and 2h). These barplots refer to the GBT models trained from Φ.

**Disentanglement.** We measure the quality of the disentanglement of the learnt models after the transfer. Since the real-world Target Datasets do not have any labels of the FoV, we evaluate the disentanglement on Texture dSprites (Source dataset) before and after the finetuning. This allows us to evaluate the "persistence" of the disentanglement after the finetuning. Target datasets do not contain all the possible combinations of their FoV and the latter do not exhibit independence, strictly required to learn disentangled representation, but with our transfer method, we expect the disentanglement to be preserved. We use MIG, DCI and the very recent OMES($\alpha = 0.5$). Figure 6 reports the OMES scores that are discussed in the next section.

**Interpretability.** We report the normalized *feature importance* of the GBT models using the Gini importance (Nembrini et al., 2018). We also provide qualitative analysis to inspect the values of the latent representations and their connection with the FoVs in the Target dataset. Figure 2 reports the features importance on the datasets.

### 3.4 CLASSIFICATION AND INTERPRETABILITY ASSESSMENT

**Lensless:** Table 1 shows the results obtained on the downstream task of plankton classification, on the disentangled representation learnt from Texture-dSprites.We compare different inputs on Ada-GVAE: (1) the original RGB images as proposed in Dapueto et al. (2024); (2) our approach with the 768-dimensional deep features Φ produced by DINO(Caron et al., 2021). The latter produces significantly higher performances. The table also compares performances before and after finetuning. Considering the huge gap between Source and Target datasets, finetuning appears to be essential, although its benefit is more evident when using Φ. The rich pretrained features provide very high results already without finetuning. To reach comparable results with disentangled features (of much smaller size, 10) requires instead finetuning.

Figures 2a and 2b show the *feature importance* before and after finetuning. We can observe that after finetuning, it may change, nicely adapting to the specificity of the dataset, where scale and texture are more relevant.

Fig. 3a reports a scatter plot of the first and second most important features (Scale and Texture) in our representation. We observe that some classes (i.e. *Actinospaerium*, *Arcella*, *Blepharisma*, *Volvox*) need just these 2 features to be clearly separable from the others, while for the others (such as *Didinium* and *Stentor*) the 2 features are less distinctive.

To provide an insight into the interpretability of the disentangled representation, we assess in Fig. 5 the Pearson correlation of the disentangled representation with a hand-crafted one obtained by

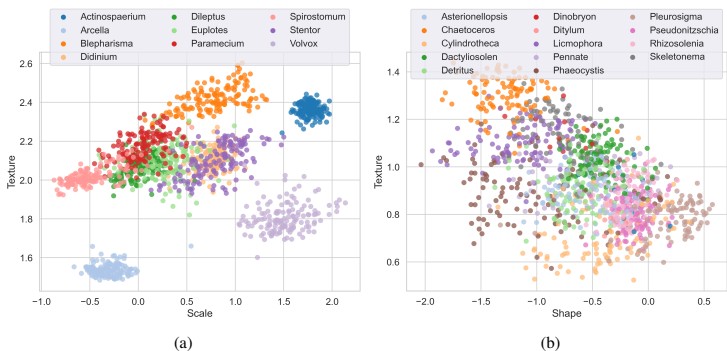

Figure 3: Representation of Lensless (Fig. 3a) and WHOI15 (Fig. 3b) using the two most important features (finetuned models with input Φ).

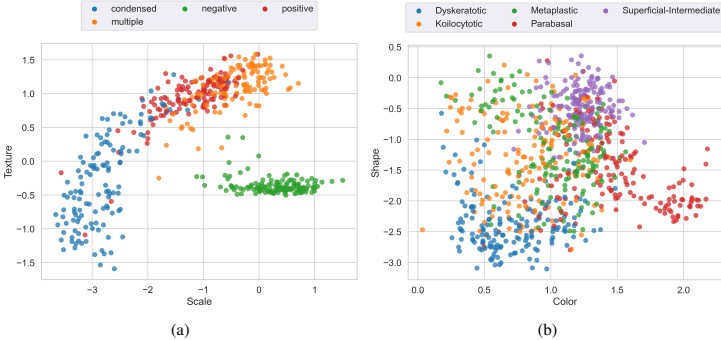

Figure 4: Representation of Vacuoles (Fig. 4a) and Sipakmed (Fig. 4b) using the two most important features (finetuned models with input Φ).

exploiting the mask ground-truth provided with the dataset to derive a set of features. Specifically, we computed a `scale` feature (as the area of the mask), `color` features (color average in the foreground) and a `shape` feature (solidity, that is the ratio of the mask area and its convex hull) (Pastore et al., 2020). Then, we computed the correlation between hand-crafted and learned (disentangled) features. For the latter, we identified the latent dimension in the disentangled representation better encoding scale, color and shape (according to the annotated source dataset). Hand-crafted and latent scale features (Fig. 5a) have a high correlation (0.86) while hand-crafted vs learnt color (we used the average red as an example in Fig. 5b) exhibit a milder correlation (-0.62). Solidity features have a smaller correlation with the dimension encoding the `Shape` factor (Fig. 5c). This might also suggest that the complexity of the shape concepts can be hardly encoded in a single (hand-crafted or learnt) value.

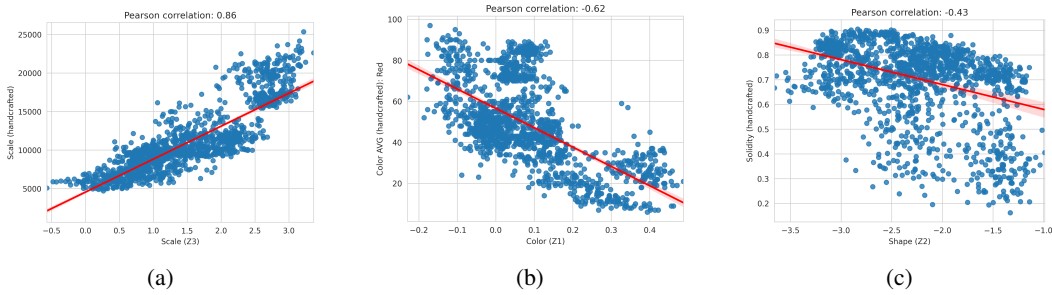

Figure 5: Correlation between our representation and handcrafted *scale* (Fig. 5a), *red channel* (Fig. 5b), and *solidity* (Fig. 5c), computed from the Lensless samples, as in (Pastore et al., 2020).

Table 2: Balanced accuracy (%) and standard deviation of the classifiers trained on the disentangled representation extracted from the VAE ( WHOI15-2007 dataset).

| S | T | Input | ✗ Finetuning | | ✓ Finetuning | |
| | | | GBT | MLP | GBT | MLP |
|---|---|---|---|---|---|---|
| T-dSpr | WHOI15 | RGB | $49.90 \pm 0.014$ | $48.20 \pm 0.018$ | $50.98 \pm 0.016$ | $49.29 \pm 0.020$ |
| T-dSpr | WHOI15 | $\Phi$ | $47.92 \pm 0.015$ | $51.96 \pm 0.023$ | $60.74 \pm 0.026$ | $63.17 \pm 0.033$ |

Table 3: Balanced accuracy (%) and standard deviation of the classifiers trained on the disentangled representation extracted from the VAE (Vacuoles dataset).

| S | Input | ✗ Finetuning | | ✓ Finetuning | |
| | | GBT | MLP | GBT | MLP |
|---|---|---|---|---|---|
| T-dSpr | RGB | $64.03 \pm 0.041$ | $59.89 \pm 0.053$ | $65.45 \pm 0.054$ | $62.77 \pm 0.057$ |
| T-dSpr | $\Phi$ | $84.95 \pm 0.02$ | $85.10 \pm 0.018$ | $90.45 \pm 0.019$ | $89.97 \pm 0.019$ |

**WHOI15-2007:** Table 2 reports the results obtained on this second, more complex, plankton dataset. We notice a significant increase in performances, as we change the input from RGB to $\Phi$ in the presence of finetuning; without finetuning, the improvement is marginal or missing because of the high intra-class and extra-class variability of the dataset that our Source dataset cannot easily capture.

Figure 2 shows the features importance before (Fig. 2c) and after finetuning (Fig. 2d). From the latter we observe that the `Color` factor is the least important considering the dataset is nearly monochromatic, while the `Texture` and `Shape` are the most important ones. Fig. 3b shows the distribution of the first and second most important features (`Texture` and `Shape`), where again we may appreciate how data are nicely clustered even for a higher number of classes.

**Vacuoles:** Table 3 reports the results on the Vacuoles dataset. Similar to the above analysis, we observe a benefit of finetuning and an improvement when deep features $\Phi$ are used. Figure 2 shows the feature importance before (Fig. 2e) and after finetuning (Fig. 2f). We observe that the FoV `Color` is the least important feature, suggesting that color (that codifies the vacuole's depth information) is the least discriminative feature for classifying the target morphotypes. Figure 4a shows the representation of `Texture` and `Scale`, we can observe that the *negative* class is aligned to the scale axis meaning the samples have the same texture but different scale, which is in line with a visual observation of the negative class in the dataset (see Fig. 10 in the Appendix).

**Sipakmed:** Table 4 shows the results obtained on the downstream task of cell classification. The analysis we can make on the results is coherent with the observations made for the previous datasets. In particular, once again we observe that the models trained on the deep feature $\Phi$ lead to a greater improvement when finetuned compared to the RGB-based one.

Comparing our results with the ones of the original work (Plissiti et al., 2018)obtained with hand-crafted features and the MLP classifier (78.92% of balanced accuracy), we observe that we achieved slightly lower performances in terms of balanced accuracy. The original work used handcrafted features concerning intensity, texture and shape calculated for both the region of the nucleus and the cytoplasm of each cell. Our learned factors take into consideration the entire image and cannot tell apart the specific information of each part of the cell. In order to improve the performances and the representation of this specific dataset, an ad-hoc Source dataset to take into account the FoVs of the separated parts of the cell would be useful. As confirmation of this, Fig. 2h shows the features' importance after the finetuning being very similar, meaning that all the features have the same importance (except for `Shape`), suggesting that ad-hoc FoVs are required to provide a more in-depth disentanglement. Figure 4b shows the representation of `Shape` and `Color`, while we observe that the *Koilocytotic* class is quite scattered and partially overlapping with most classes, *Dyskeratotic* and *Superficial* are separated. However, are all quite scattered suggesting again that this dataset may need ad-hoc FoVs.

Discussion: With an experimental analysis performed on four microscopy datasets with variable characteristics, we have shown that transferring a disentangled representation learned from a syn-

Table 4: Balanced accuracy (%) and standard deviation of the classifiers trained on the disentangled representation extracted from the VAE (Sipakmed dataset).

| S | Input | ✗ Finetuning | | ✓ Finetuning | |
|---|---|---|---|---|---|
| | | GBT | MLP | GBT | MLP |
| T-dSpr | RGB | $52.63 \pm 0.043$ | $51.25 \pm 0.050$ | $55.10 \pm 0.041$ | $55.69 \pm 0.038$ |
| T-dSpr | $\Phi$ | $61.75 \pm 0.019$ | $63.33 \pm 0.014$ | $71.17 \pm 0.025$ | $72.98 \pm 0.022$ |

thetic Source dataset to a real Target dataset is possible. We have also observed that the quality of the disentanglement and its meaningfulness for a downstream classification task may change depending on the input (RGB or deep features) and the adoption of a step of finetuning of the model to be transferred. Moreover, we compared the disentangled models obtained from raw RGB images and the deep feature $\Phi$ extracted from a pretrained network (Vit16b pretrain with DINO). We could observe that the latter allows for a more robust transfer, further improving the performances of a disentangled representation that is also human-interpretable.

However, the disentanglement of the model, which provides a level of interpretability, comes with a price. In Appendix A.2.5, we report an ablation study in which we removed the disentanglement, directly employing the deep features $\Phi$ for the downstream classification tasks. In this way, we can quantify how much we lose in terms of accuracy when advancing interpretability. We can observe that for WHOI15, the disentanglement degrades the classification performances. WHOI15 is a dataset of multi-cell images which make the data more complex and for this reason it may need further FoVs to be represented and disentangled.

### 3.5 DISENTANGLEMENT EVALUATION EXPERIMENTS

Figure 6 reports the disentanglement score for the metric OMES, that measures both *Compactness* and *Modularity*. The score without the finetuning is obtained from the original Source model trained with weak supervision, thanks to which a high level of disentanglement is obtained. The scores referring to the Target datasets are computed by extracting the representation of Texture-dSprites using the different finetuned models (reported on the x-axis) since it is not possible to do the same directly on the Target for the lack of annotation.

Overall, we observe that the models trained with the deep features $\Phi$ preserve the same level of disentanglement of the Source models independently on the Target dataset (Fig. 6a). On the other hand, the models trained with the images do not preserve the disentanglement of the Source model, and the level of degradation also depends on the Target dataset. The plots of the other metrics, MIG and DCI measuring the two properties separately, are reported in Appendix A.2.3 and our analysis is analogous for all the metrics.

To give also a qualitative result, in Appendix A.2.4 we report the scatterplot of the representation of Texture-dSprites extracted from the finetuned models, from which we can appreciate the independence of the FoV in the latent representation being alternated in the models trained on the RGB images (Fig. 15) and instead almost preserved in the models trained with the deep features $\Phi$. Moreover, in Appendix A.2.2 we show that the level of disentanglement of the original DINO features is lower than our learnt latent representation, and hence our methodology enhances disentanglement even when applied to a rich initial representation.

This suggests that **transferring from deep features extracted from pretrained models is more robust and preserves the disentanglement** also across dataset of very different domains and very different from the Source dataset.

### 3.6 PRELIMINARY ASSESSMENT ON OPEN SET CLASSIFICATION

To provide a specific application example, showing the potential of interpretability, we consider anomaly detection as a way to use plankton as a biosensor (Pastore et al., 2019; 2022; Ciranni et al., 2024). Anomalies can either correspond to novel classes or *plankton organisms reacting to environmental perturbations* (Ciranni et al., 2024). We aim to assess whether DLR can provide further information on samples detected as anomalous, allowing us to distinguish between the two described scenarios. As a study case, we remove the *Arcella Vulgaris* samples from the Lensless training set and use the remaining 9 classes for finetuning and species classification. We then feed *Arcella*

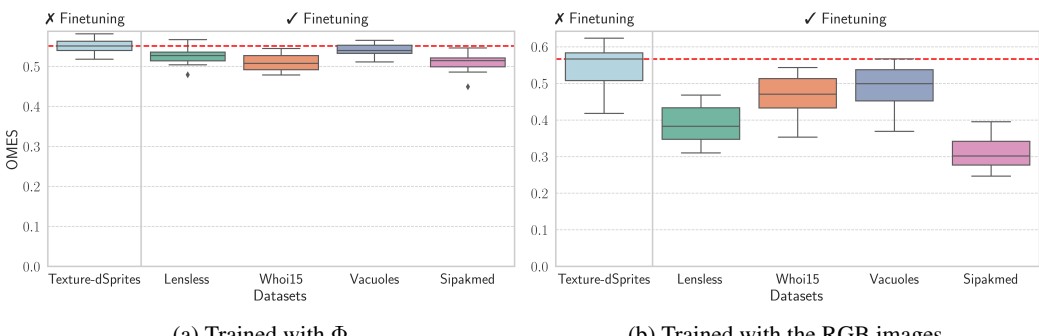

(a) Trained with Φ          (b) Trained with the RGB images

Figure 6: Disentanglement score OMES of Source and Finetuned models trained on Φ (our) or with the RGB image (Dapueto et al. (2024))

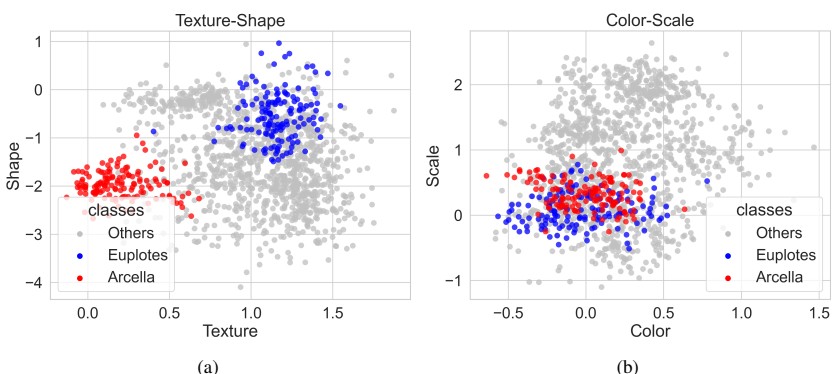

Figure 7: The representation in the `Texture-Shape` (Fig. 7a) and `Color-Scale` spaces (Fig. 7b). *Arcella* and *Euplotes* are separated or overlap depending on the features.

images to the classifier, which predicts them as *Euplotes Eurystomus*, with high confidence. We compute the mean distance for each dimension of the disentangled representation from the *Arcella* samples to the centroid of *Euplotes*, finding that `Shape` (1.42) and `Texture` (0.95) are the most further dimensions and `Color` (0.18) and `Scale` (0.27) are the closest. Fig. 7 shows how *Arcella* (red), *Eupotes* (blue), and Others (grey) samples are encoded and despite *Arcella* being classified as *Euplotes*, we can appreciate the distance between the two classes in the `Texture-Shape` space (7a), while the classes almost overlap in the `Color-Scale` space (7b). This provides insights on the actual difference between the FoVs of our test samples and the training class they are assigned to by the classifier, which could be used to identify unseen or anomalous classes in the described application scenario.

## 4 CONCLUSION

In this work, we presented a study on disentangled representation learning for microscopy images, disentangling morphological factors such as texture, color, shape, and scale. Our results on four different microscopy benchmarks suggest that the learned disentangled representations provide a good trade-off between classification accuracy and interpretability, with a finetuning protocol being particularly beneficial when deep pretrained features are used as input data.

Limitations and future work. For the time being our analysis only includes only VAE-based methods, one could carry out an analogous study with more complex and powerful methods, such as Diffusion Models. We considered a general-purpose Source dataset which may not perfectly fit the FoVs of the Target dataset, for example as observed for Sipakmed. Future work will also consider the possibility of generating a synthetic FoV annotated dataset more specific to the purpose.

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

Table 5: Texture-dSprites: the FoV generating the dataset (Left), some examples randomly generated from the FoV (Right)

## Dataset FoVs                    ## Random Samples

| Texture-dSprites | |
| --- | --- |
| **FoV** | **# values** |
| Texture | 5 |
| Color | 7 |
| Shape | 3 |
| Scale | 6 |
| Orientation | 40 |
| PosX | 32 |
| PosY | 32 |

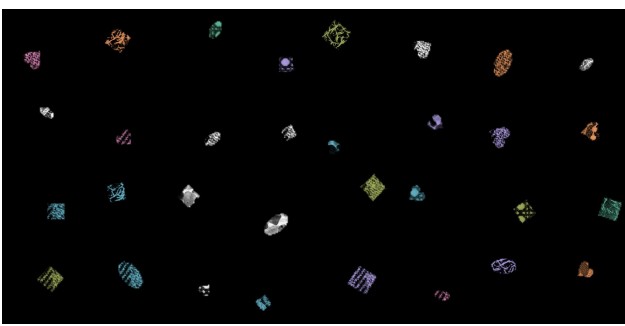

## A  APPENDIX

### A.1  DATASETS

Some samples of *Texture-dSprites* are shown in Table 5 together with the main information, i.e. Factor of Variation and number of possible values generating the entire dataset. The background is black and maintained fixed for all the samples.

We show some random examples for each class for *Lensless* Fig. 8, *WHOi15* Fig. 9, *Vacuoles* Fig. 10 and *Sipakmed* Fig. 11. We can appreciate a uniform and constant background for the Vacuoles dataset and this makes it easy to focus on the region of interest of the image. The Sipakmed dataset is characterized by a variable background and in some cases, a less defined contour of the cell. This can explain the classification accuracy reported in the results and may suggest that for such a dataset additional FoVs should be taken into account.

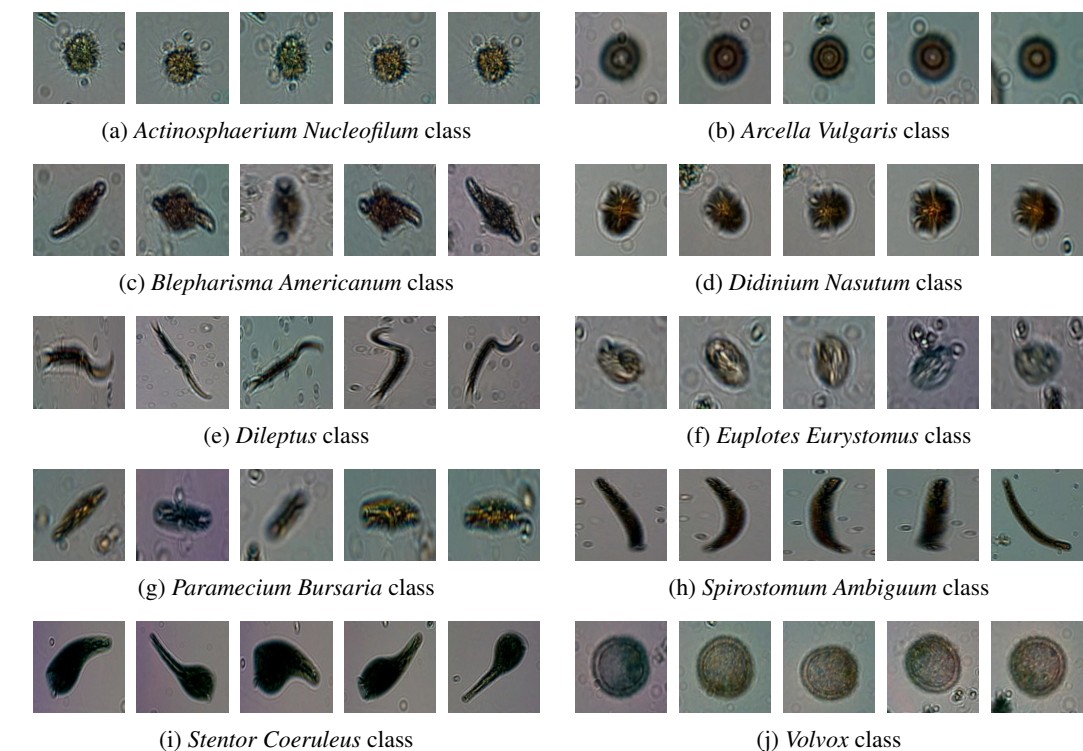

(a) *Actinosphaerium Nucleofilum* class

(b) *Arcella Vulgaris* class

(c) *Blepharisma Americanum* class

(d) *Didinium Nasutum* class

(e) *Dileptus* class

(f) *Euplotes Eurystomus* class

(g) *Paramecium Bursaria* class

(h) *Spirostomum Ambiguum* class

(i) *Stentor Coeruleus* class

(j) *Volvox* class

Figure 8: 5 random samples for each class of Lensless

Table 6: Explicitness of the FoV of Tetxure-dSprites of different pretrained networks. In **bold** the best performing model.

| Pretraining | Tetxure-dSprites FoVs (%) | | | | | | | |
| --- | --- | --- | --- | --- | --- | --- | --- | --- |
| | **Texture** | **Color** | **Shape** | **Scale** | **Orientation** | **PosX** | **PosY** | **All** |
| *Random* | 20.00 | 14.28 | 33.33 | 16.66 | 2.5 | 3.12 | 3.12 | 13.28 |
| ResNet152 | 92.50 | 96.96 | 56.93 | 53.03 | 6.17 | 16.49 | 17.52 | 48.51 |
| DenseNet201 | 96.11 | 96.78 | 77.57 | 47.67 | 8.52 | 19.09 | 19.89 | 52.23 |
| VGG19 | 97.68 | 86.24 | 87.99 | 69.87 | 19.50 | 27.49 | 38.02 | 60.97 |
| Swing4-Large | 32.80 | 21.74 | 33.39 | 21.94 | 2.50 | 8.63 | 17.21 | 19.74 |
| Vit16-Large | 99.44 | 99.98 | 93.56 | 79.56 | 20.85 | 19.35 | 17.83 | 61.51 |
| Vit16-Base + DINO | **99.74** | **100.00** | **95.36** | **88.78** | **34.00** | **31.9** | **34.74** | **69.21** |

## A.2    RESULTS

In this section, we report additional visualizations supporting the analysis discussed in the main document.

### A.2.1    BACKBONE CHOICE

We report in Table 6 the Explicitness of each FoV of Texture-dSprites obtained from the representation of different CNNs backbones (i.e. ResNet152(He et al., 2016), DenseNet201(Huang et al., 2017), VGG19(Simonyan & Zisserman, 2014)) and Transformers backbone (i.e. Swing4-Large(Liu et al., 2022b), Vit16-Large (Alexey, 2020) and Vit16-Base pretrained with DINO (Caron et al., 2021)). All pretrained on ImageNet1k.

It can be observed that ViT16-Base model pretrained with DINO self-supervised approach outperforms all the other models, hence we chose it as backbone for all the experiments.

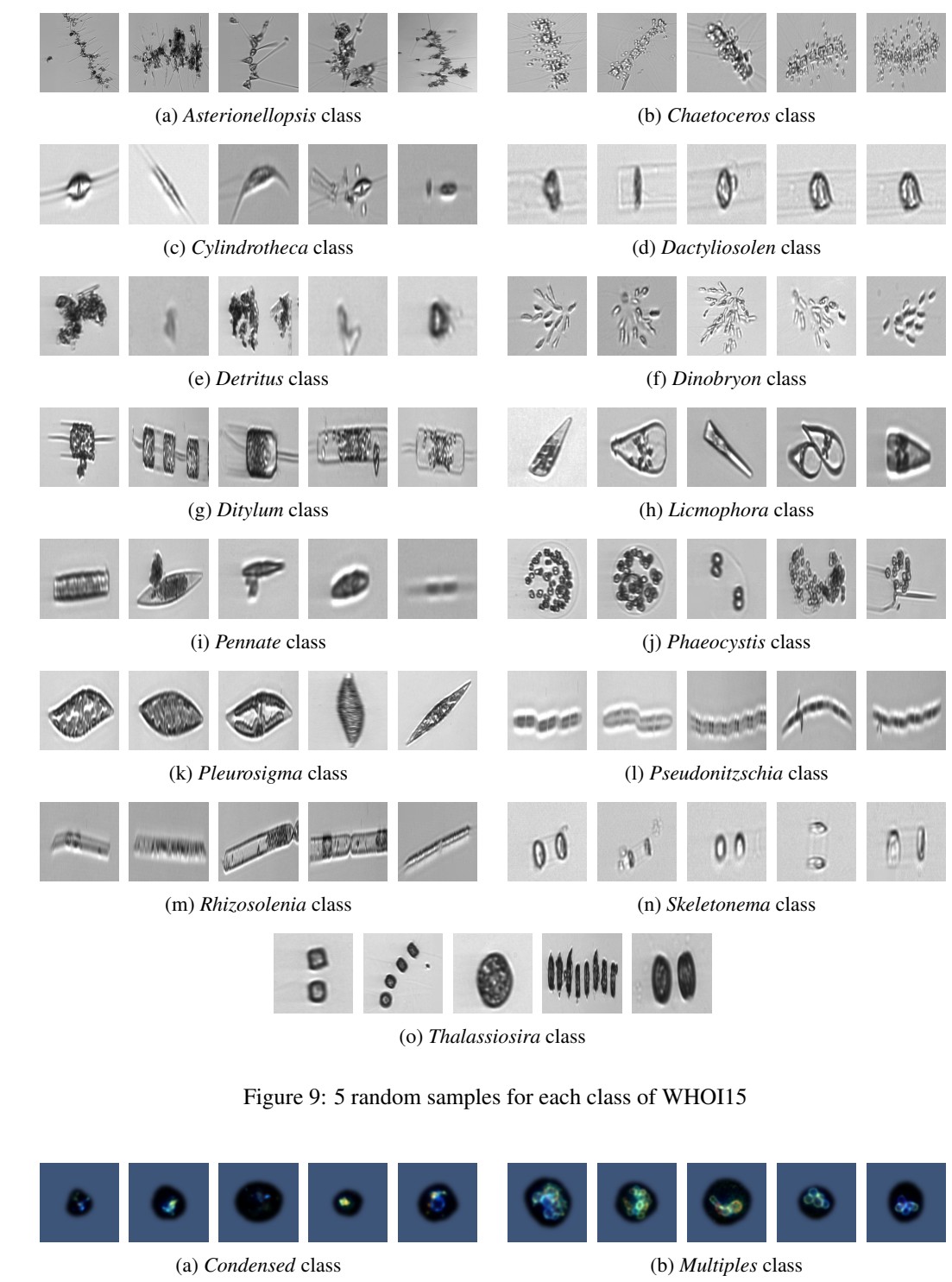

(a) *Asterionellopsis* class

(b) *Chaetoceros* class

(c) *Cylindrotheca* class

(d) *Dactyliosolen* class

(e) *Detritus* class

(f) *Dinobryon* class

(g) *Ditylum* class

(h) *Licmophora* class

(i) *Pennate* class

(j) *Phaeocystis* class

(k) *Pleurosigma* class

(l) *Pseudonitzschia* class

(m) *Rhizosolenia* class

(n) *Skeletonema* class

(o) *Thalassiosira* class

Figure 9: 5 random samples for each class of WHOI15

(a) *Condensed* class

(b) *Multiples* class

(c) *Negative* class

(d) *Positive* class

Figure 10: 5 random samples for each class of Vacuoles

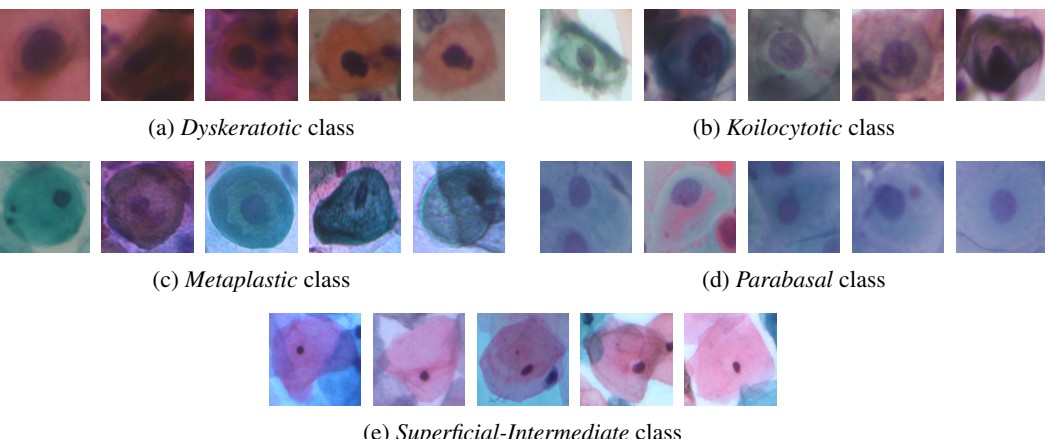

(a) *Dyskeratotic* class        (b) *Koilocytotic* class

(c) *Metaplastic* class        (d) *Parabasal* class

(e) *Superficial-Intermediate* class

Figure 11: 5 random samples for each class of Sipakmed.

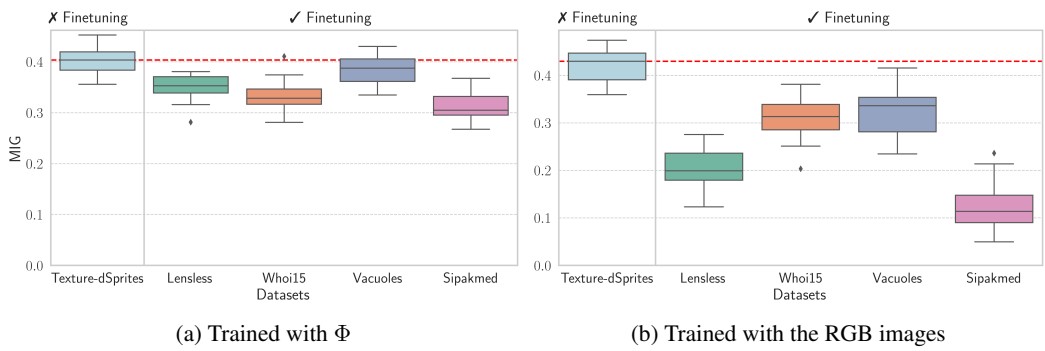

(a) Trained with $\Phi$        (b) Trained with the RGB images

Figure 12: Disentanglement score MIG of Source and Finetuned models trained on $\Phi$ (our) or with the RGB image (Dapueto et al. (2024))

Figure 4 shows the representation of the 2 most important features for the Vacuoles (Fig. 4a) and Sipakmed (Fig. 4b) datasets.

### A.2.2 DISENTANGLEMENT OF DINO FEATURES

We computed the OMES=0.26 and MIG=0.03 disentanglement scores of the DINO features, which are lower than the scores of our method in Fig.6 and Fig.12.
We did not include the metric DCI in this comparison since, as found in Cao et al. (2022), DCI does not allow a fair comparison between representations of different latent dimensions.

### A.2.3 DISENTANGLEMENT SCORE

In this section we report the disentanglement score of the Source models and the Finetuned ones. Figure 12 reports the MIG scores and Figure 13 reports the DCI scores. Analogously to what was observed for OMES in the main document, the transfer from the models trained with RGB images with finetuning leads to a degradation of the disentanglement performances that are also different from dataset to dataset. Conversely, the models trained with deep features $\Phi$ maintain similar performances to the Source model trained with weak supervision, and also the performances seem to be less dependent on the domain of the Target dataset.

### A.2.4 DISENTANGLEMENT SCATTERPLOTS

In this section, we report qualitative visualizations of the disentanglement with scatterplots of the representation of Texture-dSprites extracted from the **finetuned** models. In particular, Figure 14 and

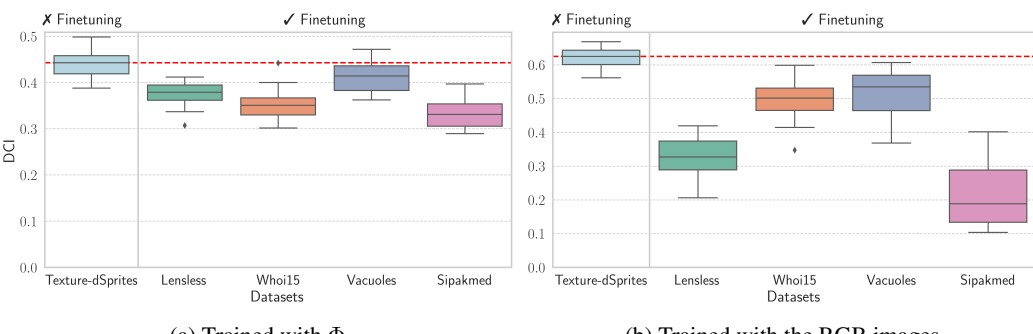

(a) Trained with $\Phi$          (b) Trained with the RGB images

Figure 13: Disentanglement score DCI of Source and Finetuned models trained on $\Phi$ (our) or with the RGB image (Dapueto et al. (2024))

Table 7: Ablation study on features removing the VAE. Balanced accuracy (%) and standard deviation over 20 classifiers.

| Dataset | Input | GBT | MLP |
|---------|-------|-----|-----|
| Lensless | $\Phi$ | $99.13 \pm 0.002$ | $99.49 \pm 1.11$ |
| WHOI15-2007 | $\Phi$ | $84.70 \pm 0.01$ | $97.16 \pm 0.01$ |
| Vacuoles | $\Phi$ | $94.52 \pm 0.01$ | $97.46 \pm 0.04$ |
| Sipakmed | $\Phi$ | $92.09 \pm 0.005$ | $94.95 \pm 0.01$ |

15 show the latent dimensions (encoding specific FoVs) and the color refers to the labels of the Fov specified in the title.

Figure 14 reports the representation extracted from the models trained on $\Phi$. Each scatter-plot visualizes 2 latent dimensions (reported on the x and y labels) while the color shade encodes the annotated values of a FoV (reported on the plot title). From the first two columns, where the FoVs are encoded in one of the latent dimensions, we can appreciate that the FoVs values are almost axis-aligned even after finetuning, meaning that navigating the latent dimension only the value of the specific FoV is changing (Horan et al., 2021). Instead, the last column represents a FoV not encoded in the 2 latent dimensions and so the values are randomly scattered. Figure 15 reports instead the representation extracted from the models trained on the original RGB images. Here we can appreciate the fact the latent representation is not so neatly agreeing with the corresponding FoV, and some of the factors are less separated than the ones in Figure 14.

A.2.5    BASELINE OF $\Phi$

In our experiments, we extract $\Phi$ from a pretrained backbone and we compress and disentangle such features by means of a VAE. In this section, we provide an empirical experiment to study the impact of the disentanglement and so providing a level of interpretability on the model prediction. So we trained the same classifiers of the previous experiment (GBT and MLP) directly on $\Phi$ and repeated for 20 random seeds, keeping the backbone weights frozen.

Table 7 show the mean balanced accuracy and the standard deviation of the classification of each target dataset. Except for WHOI15, we can observe that the impact of disentanglement on the performance is weak. WHOI15 may suffer from degradation of the performances because of information loss during the compression of the features into the latent representation $z$, and because the FoVs we are considering (Texture, Color, Shape, Scale, Orientation) does not fully capture the variability of the data. Indeed, WHOI15 may need more factors to be encoded and disentangled.

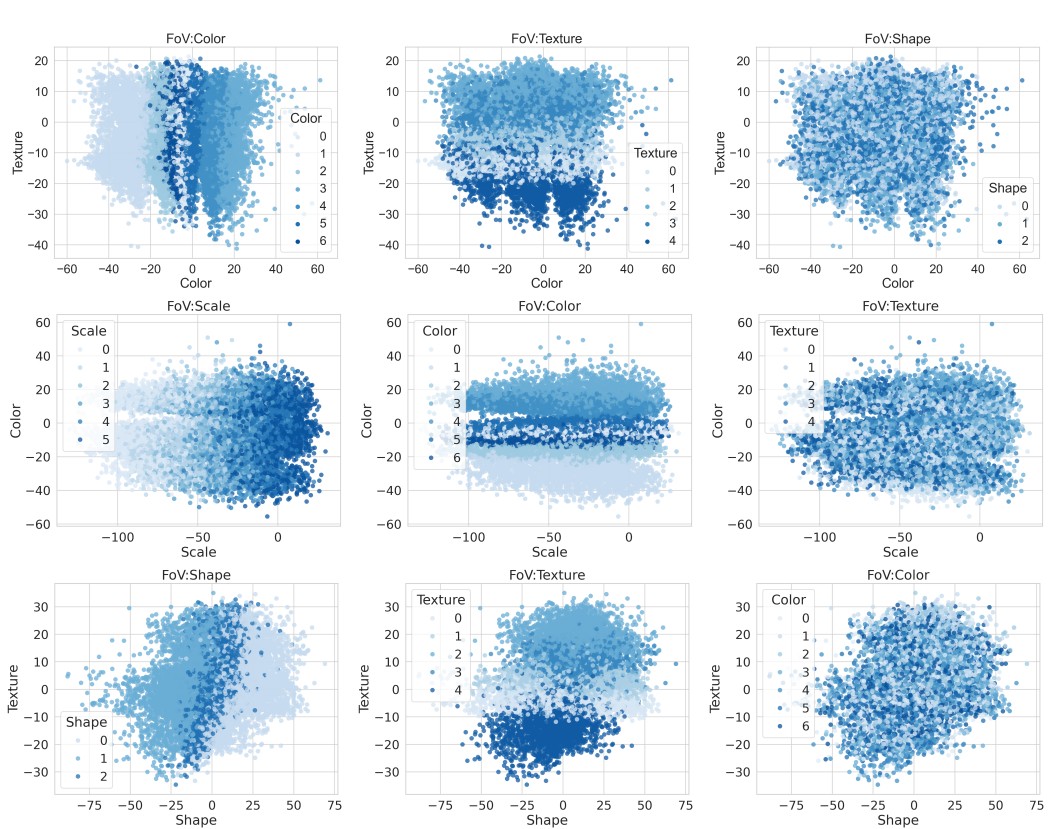

Figure 14: Representation of Texture-dSprites extracted from models trained on Φ: 2 latent dimensions at a time are considered (axes) and the color indicates the values of a given factor (reported in the title). In the first two columns the FoV is encoded in one of the two dimensions considered, while in the third column it is not, as confirmed by the fact its values are distributed randomly.

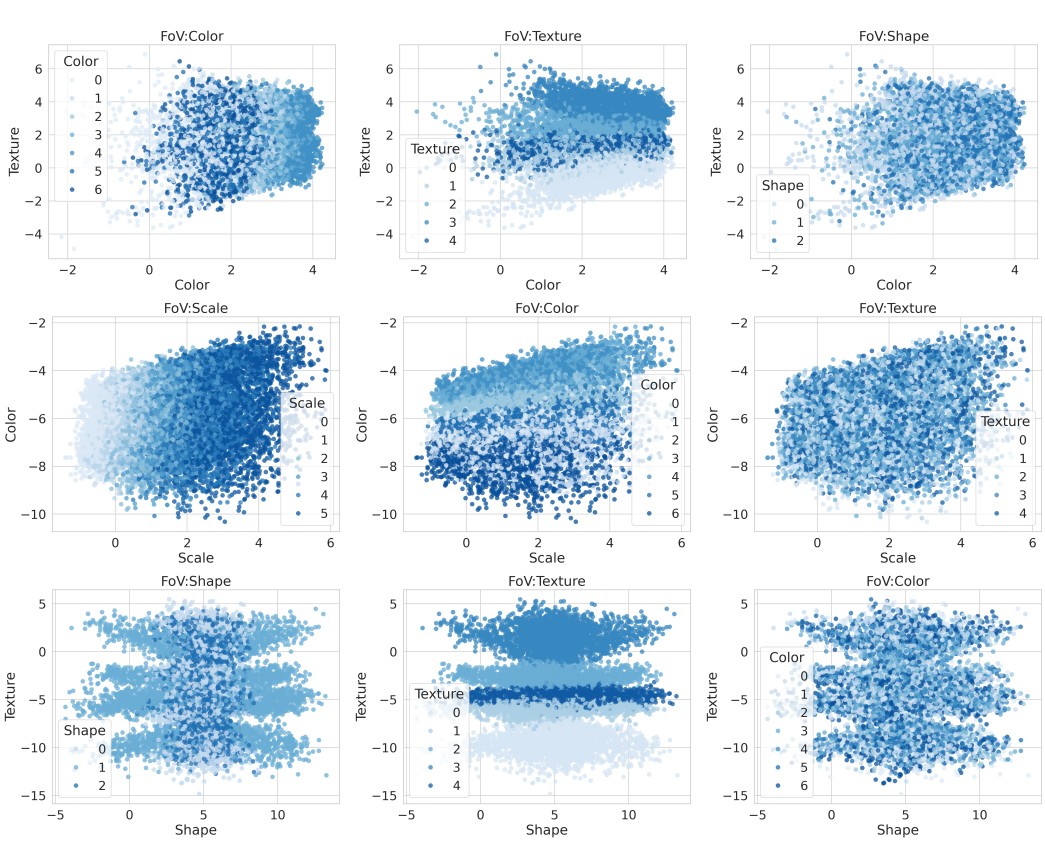

Figure 15: Representation of Texture-dSprites extracted from models trained on RGB images: 2 latent dimensions at a time are considered (axes) and the color indicates the values of a given factor (reported in the title). In the first two columns the FoV is encoded in one of the two dimensions considered, while in the third column it is not, as confirmed by the fact its values are distributed randomly.

Table 8: Encoder and Decoder architecture for the transfer experiments using the **image** as input.

| Encoder | Decoder |
|---|---|
| **Input:** $224 \times 224 \times \#$channels | **Input:** $\mathbb{R}^{10}$ |
| $4 \times 4$ *conv*, 32 LeakyRelu(0.02), stride 2 | FC 8192 LeakyRelu(0.02) |
| $4 \times 4$ *conv*, 64 LeakyRelu(0.02), stride 2 | FC $8 \times 8 \times 128$ |
| $4 \times 4$ *conv*, 128 LeakyRelu(0.02), stride 2 | $4 \times 4$ *upconv*, 64 LeakyRelu(0.02), stride 2 |
| Flatten | $4 \times 4$ *upconv*, 32 LeakyRelu(0.02), stride 2 |
| $2 \times$ FC 10 | $4 \times 4$ *upconv*, #channels Sigmoid, stride 2 |

Table 9: Encoder and Decoder architecture for the transfer experiments using the $\Phi$ as input.

| Encoder | Decoder |
|---|---|
| **Input:** 768 | **Input:** $\mathbb{R}^{10}$ |
| FC 512, LeakyRelu(0.02) | FC 1536 LeakyRelu(0.02) |
| FC 1024, LeakyRelu(0.02) | 1024, LeakyRelu(0.02) |
| 1536, LeakyRelu(0.02) | 512, LeakyRelu(0.02) |
| $2 \times$ FC 10 | FC 768 , Sigmoid |

## A.3 ARCHITECTURE

Table 8 shows the architecture of all the models trained with images, while Table 9 shows the architecture of all models trained with the deep features $\Phi$.

## A.4 COMPUTING RESOURCES

All the experiments have been executed with an NVIDIA Quadro RTX 6000. On average, the training and evaluation of a single Source model take 6 hours. Each finetuning and final evaluation takes 2.5 hours. Overall, the whole bunch of transfer experiments and our metric assessment took approximately 500 hours.

