# OpenReview forum: "Disentangled representations of microscopy images"
_ICLR.cc/2025/Conference — Submitted to ICLR 2025_

### Official Review · Reviewer_ijPz · 2024-10-29

**Soundness:** 1
**Presentation:** 2
**Contribution:** 1
**Rating:** 3
**Confidence:** 4

**Summary:**

In this paper, the authors propose to use a disentangled representation learning framework to enhence model interpretability for microscopy image classifications. The method is based on fine-tuning a model trained on synthetic images, the proposed framework is tested on some microscopy images datasets.

**Strengths:**

- The paper addresses a significant challenge in representation learning: disentanglement, which plays a pivotal role in improving the interpretability of classifiers, particularly in the context of biological images.

**Weaknesses:**

- The proposed approach is not well explained. Indeed, the method proposed by the authors learns a disentangled model with weak-supervision using Ada-GVAE on a synthetic dataset and then fine-tune it on
 a microscopy datasets. However, it is unclear why Ada-GVAE is choosed and how is the model fine-tuned.

- The difference between the proposed method and Dapueto et al is unclear.

- The authors claim that the disentanglment learned from a synthetic images can be transferred to microscopy images, such claim should be theoretically and empirically evidenced.

- The paper is not well organized, for instance a "Related Work" section should be added. Two different sections (2.2 and 3.5) have the same title "DISENTANGLEMENT EVALUATION".

**Questions:**

- In Fig, 1, what is exactely fine-tuned and how ?

- How is an RGB image directly fed to the classifier (GBT and MLP)?

- In line 322, the authors state "We can observe that after finetuning, it may change, nicely adapting to the specificity of the dataset, where scale and texture are more relevant.", It is unclear for me why scale and texture are more relevent then "scale and shape", as it is the case before fine-tuning.

- The proposed evaluation metrics (ex:OMES) are unclear.

- The authors do not compare their method to any other work, having a solid baseline is important.

- The used classifiers (GBT and MLP)  are very simple, more sophisticated ones should be used (CNNs based for example).

- Inputting an RGB image to the classifer is unclear as it is well-established that deep features (in this cas the features extracted by DINO) have more important patterns.

- To assess the quality of the representation, the authors realied on classification. While a good representation can lead to a better accuracy. A good representation does not necessary mean a disentangled one.

- Using the accuracy only to measure the quality classification performance is not enough.

- The figures are small and the captions are not clear enough.

- In Figure 6, the OMES indicates that the proposed method does not lead to better disentanglement.

---

> ### Author Response · Authors · 2024-11-25
>
> We thank the reviewer for their comments. In the following we answer each concern separately.
>
> * > The proposed approach is not well explained ...  how is the model fine-tuned.
>
> We rely on weakly supervised methods that require less annotation to be trained. Moreover, we chose VAE-based models because of their simplicity which allows us to analyze more intuitively the effectiveness of the disentanglement. Our hypothesis is that the observations made on VAEs can generalize to alternative DLR methods, as for instance [1, 2, 3], more complex and hence powerful than the simple VAEs.
> Future work will be devoted to empirically assess this hypothesis.
>
> As for the fine-tuning, once the (weakly-supervised) Ada-GVAE model has been trained on the Source dataset, it is finetuned on the (unsupervised) Target using a Beta-VAE model.
> We will clarify it in the main document.
>
> * > The difference between ... is unclear.
>
> Please, see the general answer
>
> * > The authors claim that the disentanglment ... and empirically evidenced.
>
> The theoretical transferability of disentangled representation learnt from synthetic images has been proved in [4]. Empirically showing that we can transfer DR to real more complex data is the goal of this work.
>
> * > The paper is not well organized ... "DISENTANGLEMENT EVALUATION".
>
> Concerning the related works, it is true that a dedicated section is missing, but this is because we opted to provide an account of the relevant literature in the introduction.
> On the second issue, the reviewer is right, we will rename the Sec. 2.2 “DISENTANGLEMENT EVALUATION METHODS” and Sec. 3.5 as “DISENTANGLEMENT EVALUATION EXPERIMENTS”
>
> * > In Fig, 1, what is exactely fine-tuned and how ?
>
> It is fine-tuned as a Beta-VAE model [5]. In the old version of Fig. 1 we identified the frozen modules with the icon of a snowflake. To be more clear, in the new version of the figure, we also marked the modules involved in the finetuning with a flame.
>
> * > How is an RGB image directly fed to the classifier (GBT and MLP)?
>
> The RGB images are not fed directly to the classifiers. We first fed the RGB image to the VAE encoder extracting the representation z, we then trained the classifiers using z, as shown in Fig.1. More details can be found in Sec.3.3 .
>
> * > The proposed evaluation ... are unclear.
>
> We adopted the MIG and DIC disentanglement evaluation metrics because they are the most used in previous work, while OMES is a very recent metric that has been shown to be more robust across datasets of different natures, while providing interpretability of the results and ensuring a reliable assessment of DR properties.
>
> * > The authors do not compare ... baseline is important.
>
> See the general answer
>
> * > The used classifiers (GBT and MLP) ... (CNNs based for example).
>
> As we reported in the paper, Sec. 2.2 “As for the choice of the classifier, a general criterium is to select a simple model, to better appreciate the influence of the representation on the performance in the downstream task.“ In other words, since our objective was to assess the power of the DR, we needed the classifiers to be simple not to influence the results. On the specific choices, we referred to [6].
>
> * > To assess the quality of the representation ... a disentangled one.
>
> While this is true, our work aims to learn a disentangled representation which not necessarily obtains the best result on classification. It is part of the well-known, trade-off between performance and interpretability.
> More classical methods for classifying biological data rely on identifying and extracting handcrafted features of the images which could indeed resemble the FoVs one may identify in the datasets. In this sense, our work aims to learn the FoVs of the microscopy data following a fully data-driven approach instead.
>
> * > The figures are small and the captions are not clear enough.
>
> We thank the reviewer for the observation. We will make more descriptive captions.
>
> * > In Figure 6, the OMES ... to better disentanglement.
>
> We thank the reviewer for the observation.
> We noticed a mistake in the figure, which is now solved in the new version. Correcting the problem, the OMES score of the proposed method is now comparable to the ones using the RGB images.
>
> [1] Yang, et al. "Disdiff: Unsupervised disentanglement of diffusion probabilistic models.", in NeurIPS 23.
>
> [2] Song, Yue, et al. “Flow Factorized Representation Learning.”, in NeurIPS 2023.
>
> [3] Lin et al., “Infogan-cr and modelcentrality: Self-supervised model training and selection for disentangling gans”, In ICML 20.
>
> [4] Dapueto et al. "Transferring disentangled representations: bridging the gap between synthetic and real images", arXiv:2409.18017: NeurIPS 2024 (accepted)
>
> [5] Higgins, Irina, et al. "beta-vae: Learning basic visual concepts with a constrained variational framework.", in ICLR 2017.
>
> [6] Dittadi, Andrea, et al. "On the Transfer of Disentangled Representations in Realistic Settings.", ICLR 2021.

---

> ### Comment · Reviewer_ijPz · 2024-12-02
> **Official Comment by Reviewer ijPz**
>
> I thank the authors for their response. Unfortunately, many of my concerns were not addressed. For this reason, I will maintain my current scores.

---

> > ### Author Response · Authors · 2024-12-02
> >
> > Thank you for your time in considering our response. Could you be a bit more specific about what concerns we haven’t addressed?

---

> > > ### Comment · Reviewer_ijPz · 2024-12-02
> > > **Official Comment by Reviewer ijPz**
> > >
> > > - The novelty of the work remains unclear to me. In my opinion, this study appears to be an application of [1] to certain microscopy datasets.
> > >
> > > - The evaluation metrics are not clearly explained. Using classification to assess the quality of disentanglement lacks clarity.
> > >
> > > - I encourage the authors to adapt existing methods to establish a baseline for comparing the performance of their model.
> > >
> > > [1] Dapueto et al. "Transferring disentangled representations: bridging the gap between synthetic and real images", arXiv:2409.18017: NeurIPS 2024 (accepted)

---

> > > > ### Author Response · Authors · 2024-12-03
> > > >
> > > > *  >The novelty of the work remains unclear to me. In my opinion, this study appears to be an application of [1] to certain microscopy datasets.
> > > >
> > > > The comment of the reviewer may unveil the opinion that the level of novelty of our work is not sufficient for ICLR. However, we respectfully believe that a good level of novelty is present in our work, as we explained in the first part of the general response.
> > > >
> > > >
> > > > * > The evaluation metrics are not clearly explained. Using classification to assess the quality of disentanglement lacks clarity.
> > > >
> > > > We apologize if this part is unclear. As specified in response to another reviewer, unfortunately, it is not possible to compactly describe the metrics with simple formulas, as they are quite structured algorithms (especially OMES and DCI). The choice of using DCI and MIG is due to their popularity in previous works, while OMES is used in Dapueto et al., which is a reference work for our method. While providing all the details in the main paper is not possible because of space constraints, we highlighted what we believe are the minimal yet essential information to understand and interpret the metrics, i.e. the algorithmic principles they rely on, and the properties of disentangled representations they can capture (see Sec. 2.2).
> > > > 	In this answer, we try to be more specific, having space for all the clarifications. The MIG and DCI metrics are well-known in the field of disentanglement learning, and designed to quantify the intensity of one or more particular properties of a disentangled representation: DCI measures the *Modularity*, i.e. how much the FoVs 	affect non-overlapping partitions of the representation; MIG measures the *Compactness*, related to the size of the representation space affected by a FoV, that should be as small as possible; OMES measures both Modularity and Compactness in an unified manner.
> > > > 	From the point of view of the methodology, the metrics follow different approaches. In particular, DCI is based on a learnable regressor (e.g. Decision Tree) to score the importance of each latent dimension to each FoV, and different factors should be important for different dimensions. MIG is based on a Mutual Information 	Estimator and computes the mutual information between the factors and the partitions of the representation. OMES makes use of the correlation to build an association matrix (factor, dimension) to compute both Modularity and Compactness from it. Concerning the classification tasks, the assessment serves to verify if the 	FoVs encoded in the representation are enough to describe the dataset and useful for a downstream task. Furthermore, in this field, the classification score is commonly used to assess another desired property for disentangled representations, i.e. *Explicitness*.
> > > >
> > > > * > I encourage the authors to adapt existing methods to establish a baseline for comparing the performance of their model.
> > > >
> > > > we are not sure we fully understand the suggestion. If the reviewer is referring to a baseline for disentanglement learning, then we emphasize that Dapueto et al. is the baseline for our work, and a method we compare with. If instead, the reviewer is referring to a classification baseline, then the comparison is the Appendix (Table 7). On this, we highlight that our work aims to learn a disentangled representation which not necessarily obtains the best result on classification. It is part of the well-known, trade-off between performance and interpretability.

---

### Official Review · Reviewer_MZMx · 2024-11-01

**Soundness:** 1
**Presentation:** 1
**Contribution:** 2
**Rating:** 1
**Confidence:** 5

**Summary:**

The paper proposes a Disentangled Representation Learning (DRL) approach to improve interpretability in microscopy image classification. By pre-training on synthetic data (Texture-dSprite) to capture factors of variation, the authors apply these learned representations to real-world microscopy datasets (Plankton Lensless, Plankton WHOI15, Budding Yeast Vacuoles, and Sipakmed Human Cells). Their method aims to support model interpretability while achieving high classification performance with gradient-boosted trees and MLPs for downstream analysis.

**Strengths:**

The paper explores the application of an existing DRL framework to the specific domain of microscopy images. This idea is interesting as it shows a potential pathway for combining DRL with microscopy image analysis.

**Weaknesses:**

A significant weakness as it seems, is the absence of a comparison with other similar methods. The paper presents only one framework and does not discuss or evaluate alternative approaches, which weakens the case for this framework’s efficacy or advantage over existing methods.

The contributions of the paper in terms of novelty are unclear. The study applies an existing DRL approach to a new domain but does not appear to introduce any fundamentally new concepts, techniques, or substantial modifications to existing methods. The only apparent novelty - the application of DRL to microscopy imaging does not suffice. This limits the potential impact and originality of the work.

The paper’s presentation suffers from numerous issues that impede readability and clarity:
1. There are instances of informal languages, such as the use of “thanks.”
2. The text contains multiple errors at the word, sentence, and structural levels, which disrupts the reading experience. Sections like Section 2.2 (“Disentanglement Evaluation”) resemble output generated by ChatGPT and lack rigorous academic polish.
3. Figures appear low-resolution, with inadequate explanations in captions. Captions should be comprehensive and self-contained, but here, they lack essential details, e.g., explanations of metrics like OMES and balanced accuracy.
4. The use of multiple highlight types (underscoring, bold, italics) is excessive and distractive. Minimal highlighting would improve readability and make essential points more accessible.
5. Important metrics are either not explained in the text or lack adequate definitions in the captions, leaving readers uncertain of their meaning. This omission impacts the study’s reproducibility and overall clarity.

**Questions:**

Most of my questions are related to major weaknesses.

What specific contributions does this paper make beyond applying DRL to microscopy images? It would be helpful if the authors could clarify what is novel in their approach and how it advances the state-of-the-art in microscopy image analysis beyond existing techniques.

What are alternative approaches the authors could have used for comparison?

Metric explanations (e.g., OMES, MIG, DCI and balanced accuracy) are mostly missing. Could the authors clarify these metrics, ideally using mathematical notation and provide justification for using them?

---

> ### Author Response · Authors · 2024-11-25
>
> We thank the reviewer for their observations. In the following we answer each concern separately.
>
>
> * > A significant weakness as it seems ... over existing methods.
>
> Please, see the general answer
>
> * > The contributions of the paper in terms of novelty are unclear ... and originality of the work.
>
> See the general answer
>
>
> * > 1. There are instances of informal languages, such as the use of “thanks.”
>
> We thank the reviewer for the observation. We will carefully revise the style of the language.
>
> * > 2. The text contains multiple errors at the word ... rigorous academic polish.
>
> We respectfully reject any accusation of having used ChatGPT inappropriately. However, we know that Sec. 2.2 is quite dense with citations that may affect readability: we will move citations at the end of the sentences and revise the section.
>
> * > 3. Figures appear low-resolution ... balanced accuracy.
>
> On the quality of the pictures, we checked and it is due to a “compression” in OpenReview which is out of our control (the paper we submitted does not have this issue). We will carefully revise the text of the captions to ensure all the necessary details are present.
>
> * > 4. The use of multiple highlight types ... more accessible.
>
> We thank the reviewer for the observation. We will reduce the use of highlighting to improve readability.
>
> * > 5. Important metrics are ... and overall clarity.
>
> Unfortunately, it is not possible to compactly describe the metrics with simple formulas, as they are quite structured algorithms (especially OMES and DCI). We provided the code as supplementary material to allow the work to be reproduced. We will enrich the captions with the necessary details for readability.
>
> * > What specific contributions does ... beyond existing techniques.
>
> See the general answer.
>
> * > What are alternative approaches the authors could have used for comparison?
>
> See the general answer

---

### Official Review · Reviewer_umB6 · 2024-11-03

**Soundness:** 2
**Presentation:** 2
**Contribution:** 1
**Rating:** 3
**Confidence:** 3

**Summary:**

This paper addresses the interpretability challenge in microscopy image analysis using deep learning approaches. The authors propose a novel methodology based on Disentangled Representation Learning (DRL) to enhance model interpretability while maintaining classification performance. The approach leverages transfer learning from synthetic features and is validated across three diverse microscopy domains: plankton, yeast vacuoles, and human cells. The growing volume of microscopy images due to technological advances has necessitated automated analysis methods, yet interpretability remains crucial for practical applications in fields such as diagnosis and environmental monitoring. The authors demonstrate that their DRL framework successfully balances the trade-off between model accuracy and interpretability in microscopy image classification tasks.

**Strengths:**

1. The manuscript is well-written and easy to follow, with clear organization and logical flow.
2. The application of weakly-supervised DRL to real-world image analysis represents a promising and valuable research direction.

**Weaknesses:**

1.The scope of this work appears too narrow, focusing solely on microscopy images.  The proposed approach might be more convincing if demonstrated on natural images as well.
2.The authors fail to adequately justify why DRL should be specifically applied to microscopy image analysis.  Furthermore, they do not clearly articulate whether this specific application domain poses new challenges or requirements for DRL that could lead to innovative solutions.  The authors' insights into these aspects are not well presented.
3.Given the lack of compelling insights, this work appears to be primarily an application of existing DRL methods without significant methodological or theoretical innovation. This level of contribution may not align with ICLR's focus on novel methodological and theoretical advances in machine learning.
4.The paper appears to lack comparative experiments. While the disentanglement scores might be novel evaluation metrics, the absence of comparisons for classification performance is particularly concerning and unreasonable.

**Questions:**

Referring to the weaknesses noted above, I find the claimed contributions of this paper not sufficiently convincing.  Could the authors provide a more compelling explanation of their main contributions, particularly addressing:
1. Why DRL is specifically suited for microscopy image analysis.
2. What novel challenges or requirements this domain brings to DRL.
3. How their approach advances the theoretical or methodological aspects of DRL beyond simple application.

---

> ### Author Response · Authors · 2024-11-25
>
> We thank the reviewer for their comments. In the following we answer each concern separately.
>
> * > 1.The scope of this work ... natural images as well. 2. The authors fail to ...  are not well presented.
>
> Please, see the general answer.
>
> * > 3.Given the lack of compelling insights ... advances in machine learning.
>
> See the general answer
>
> * > 4.The paper appears to ... concerning and unreasonable.
>
> See the general answer
>
> * > 1.Why DRL is specifically... 2. What novel challenges...3. How their approach advances...beyond simple application.
>
> While these issues have been addressed more extensively in the general answer, we also provide a summary here:
> 1. From the methodological point of view, applying DRL directly to Biological data is unfeasible because of lack of FoVs annotation, their dependence, etc. We study the application of the transfer methodology to Microscopy datasets because they exhibit the challenges of the real data while keeping the complexity of FoVs under control. We remark that DRL has been hardly applied to real data to disentangle most of the FoVs to get a finer and complete representation of real data.
> 2. Our advancement is not on theoretical aspects of DRL but is on the application of DRL to real images, which has been underexplored so far. In this sense, we believe the work to be of interest for the ICLR conference, where among the topics of interest in the call for papers we find the applications to physical sciences (including biology).

---

### Official Review · Reviewer_MkND · 2024-11-03

**Soundness:** 2
**Presentation:** 2
**Contribution:** 1
**Rating:** 3
**Confidence:** 5

**Summary:**

The paper presents a study of disentangled representation learning on three microscopy image datasets. The representation learning strategy starts by training an Ada-GVAE model using a Textures-dSprites dataset introduced in this work. The dataset is supposed to reflect simple textures that could help interpret information in microscopy images. After training this model in a weakly supervised way, it is used to encode images of another domain, with optional unsupervised finetuning using a beta-VAE. The resulting features are low dimensional, and interpretable, and are used to train classifiers.

The ideas and the study are generally interesting, but the paper lacks technical novelty, is limited to a small-scale empirical evaluation only, and the experiments are incomplete to fully understand the value of the proposed strategy.

**Strengths:**

* The paper evaluates the recent ideas of disentangled representation learning using weak supervision in a more realistic application.
* The paper also presents an alternative to learning the disentangled representation from RGB images based on models pretrained at large scale.
* The paper proposes a new sprites dataset to facilitate the interpretation of microscopy images.

**Weaknesses:**

* The technical contribution is limited. Beyond the sprites dataset and the use of pretrained features, many of the ideas have been presented in previous works.
* The experimental evaluation is limited to quantifying the impact of classifier types (GBT vs MLP) and input type (RGB vs DINO features). Many questions remain open regarding how much classification accuracy could be obtained without the proposed disentanglement procedure. Can the authors compare results of training a classifier directly with RGB images and another classifier with DINO features without any modifications? These results would help understand how difficult the tasks are and what is the trade-off between using disentanglement vs not using it.
* It is possible that DINO features are already disentangled and all what the proposed strategy is doing is assigning names to some of the factors of variation that DINO can detect. Therefore, the disentanglement is not really happening in the VAEs but rather obtained from a model pretrained at large scale. What type of experiment can the authors design to test this hypothesis?
* If the hypothesis above is not rejected, the value of proposed methods is limited to only annotating factors of variation rather than identifying them in a weakly supervised manner to then being transferred.

**Questions:**

Can the authors clarify the questions above? Specifically, the extent to which DINO already offers certain degree of disentanglement and how the factors of variation of interest could be identified directly from these representations.

---

> ### Author Response · Authors · 2024-11-25
>
> We thank the reviewer for their comments. In the following we answer each concern separately.
>
> * > The technical contribution is limited...previous works
>
> Please, see the general answer
>
> * > The experimental evaluation is limited ...  not using it.
>
> The classification accuracy without the proposed disentanglement can be found in Appendix A.2.4. In particular, we report the accuracy of the GBT and MLP classifiers trained with the DINO features without disentanglement. We notice the classification performance is higher than the one obtained with the disentangled representation. However, we highlight the fact that the representation in that case is much larger than the disentangled one (768 vs 10 features).
>
>
> * > It is possible that DINO features ... to test this hypothesis?
>
> To test this hypothesis, we perform a new experiment, computing two disentanglement scores (OMES, MIG) of the DINO features w.r.t. the FoVs (Texture, Color, Shape, etc.) of Texture-dSprites. By employing the whole DINO feature vector (size 768) we get OMES=0.26 and MIG=0.03 .
> As shown in Fig. 6, for our disentangled features OMES=0.55, and MIG=0.4 (Fig. 11). This experiment confirms that disentanglement does not come naturally from DINO features, while it is obtained thanks to the VAE and our proposed strategy.
> We specify that we did not include the metric DCI in this comparison (which we use in the paper) since it is purely based on a classifier: when the description is very compact, as in common DR, a good classification performance is a sign of robustness and explicitness, a desired property of DR; however, for larger descriptors (as in the case of DINO features) a high classification score is not a sign of high disentanglement, but rather of reliability of the pre-trained features, which we expect. Moreover, as found in [1], DCI does not make it possible a fair comparison between representations of different latent dimensions.
> Nonetheless, we highlight that the usage of pre-trained deep features is one of the key contributions of this work, allowing a good transfer between synthetic FoV’s and real datasets, as supported by the improvement in accuracy for our disentangled features with respect to [2] (only employing images, and as such, corresponding to our column Images in Tables 1, 2, 3 and 4).
>
> [1] Cao, Jinkun, et al. "An empirical study on disentanglement of negative-free contrastive learning.", in NeurIPS 2022
>
> [2] Dapueto et al. "Transferring disentangled representations: bridging the gap between synthetic and real images", arXiv:2409.18017: NeurIPS 2024 (accepted)

---

> > ### Comment · Reviewer_MkND · 2024-11-27
> >
> > Thanks to the authors for their comments and clarifications. After careful consideration of the manuscript, the responses, and the comments by other reviewers, I will keep the current score unchanged.

---

### Author Response · Authors · 2024-11-25
**General answer**

We thank all reviewers for their valuable comments and suggestions. In this part, we address issues of interest for more than one reviewer.

* Rev. **MkND**, Rev. **umB6** and Rev. **MZMx** express concern about *Technical novelty and contributions*. \
We start by highlighting the fact we are submitting the work to the area **applications to physical sciences (physics, chemistry, biology, etc.)**: in this work, we target the advancement of the applicability of DRL to real images to obtain an interpretable and reliable representation. We emphasize different contributions in our work related to methodology and application.\
The *first* level of novelty, which we believe is not purely technical, is to perform DRL relying on deep pre-trained features. Specifically, our results show how using ImageNet to pre-train a vision transformer, in a self-supervised framework, allows for a good FoVs transfer from a synthetic dataset (Texture-dSprite) to three different microscopy datasets. The usage of deep pre-trained features is fundamental for supporting the FoVs transfer, as highlighted by better performance w.r.t. [A] (that only employs images) on the investigated datasets. \
A *second* level of novelty is the use of real data for DRL.  To the best of our knowledge, no studies about fine-grained DRL of FoVs in real natural images exist, because of real FoVs dependence, resolution, and variability, bringing the level of complexity of the problem to another scale. Besides, there is also the need for annotated data, that precisely for the same reasons is a complex and ambiguous procedure, since sometimes the FoVs can not even be clearly identified [D]. Previous works (e.g. [B, C]) attempt to transfer disentangled representation targeting either synthetic or simulated data. A first advance in the use of real data can be found in [A], where the authors analyzed the transferability of disentanglement (learned on a Source synthetic dataset) to a real Target dataset where FoVs are explicitly known. However, in this existing work, the authors adopt real data of controlled complexity.
In this work, we aim to move a step further, studying the applicability of DRL (and specifically, transferring FoVs learned with appropriate synthetic datasets) to the microscopy image domain, where  FoVs are only partially known. \
Finally, we want to provide a justification on the choice of the specific application domain, which may appear narrow, but actually presents different interesting challenges for DRL. Indeed, microscopy single-cell images have been selected for other two different reasons: \
(i) The first one is methodological, since these real images have only partially known FoVs,  representing a perfect test field for advancing towards the DRL application to general natural images; \
(ii) The second one is application-oriented: microscopy image applications are in strong need of interpretability measures, especially considering the clinical domain, where having the possibility to analyze inferred FoVs, while maintaining a reasonable accuracy, may improve the trust in AI and its employment in real-world related applications.

* Rev. **MZMx**, Rev. **umB6**  and Rev. **ijPz** express concern about *Comparison with other methods* for classification and DRL. \
It is worth noting that our primary aim is to obtain a disentangled representation of microscopy images, it should be also useful for downstream tasks. For this reason, we assessed the representations on classification tasks for each dataset, even though achieving the best classification performance is not our main goal. \
In the literature, DRL applications to real-world datasets are very limited. In this sense, to the best of our knowledge, the only suitable work for comparison corresponds to [A], that we used “as it is” when using images in input to our methodology (indeed this gives the comparison, we will better clarify this in the revised version of the manuscript). \
Specifically, [A] study the transferability of disentanglement to real data constraining some of the challenges typical of real images for DRL. In this sense, it is very appropriate for our purposes. A further element of complexity in our work is the unavailability of the FoVs' annotations on the real target datasets.

Finally, in the revised paper, we will clarify the scope of the work in the introduction and contributions while adding comparative classification results from the literature.

[A] Dapueto et al. "Transferring disentangled representations: bridging the gap between synthetic and real images", arXiv:2409.18017: NeurIPS 2024 (accepted)

[B] Gondal, Muhammad Waleed, et al. "On the transfer of inductive bias from simulation to the real world: a new disentanglement dataset.", NeurIPS 2019.

[C] Dittadi, Andrea, et al. "On the Transfer of Disentangled Representations in Realistic Settings.", ICLR 2021.

[D] Xiang et al “Disunknown: Distilling unknown factors for disentanglement learning”, ICCV 2021

---

### Meta-Review · Area_Chair_orkH · 2024-12-20

**Metareview:**

The paper explores the application of existing methods for learning disentangled representations to the problem of interpretable image classification in biological microscopy image analysis. The reviewers appreciated the application but unanimously recommended rejection citing limited technical novelty.

**Additional Comments On Reviewer Discussion:**

The reviewers and authors engaged in sufficient back and forth during the discussion period, but ultimately the major weaknesses in the paper could not be addressed.

---

### Decision · Program_Chairs · 2025-01-22

Reject